# The Moderating Role of Self-Care Behaviors in Personal Care Aides of Older Adults during the COVID-19 Pandemic

**DOI:** 10.3390/ijerph20065177

**Published:** 2023-03-15

**Authors:** M. Graça Pereira, Ana Filipa Gonçalves, Laura Brito

**Affiliations:** 1Psychology Research Centre, School of Psychology, University of Minho, 4710-057 Braga, Portugal; 2School of Psychology, University of Minho, 4710-057 Braga, Portugal

**Keywords:** self-care, formal caregivers, psychological morbidity, COVID-19 preventive behaviors, COVID-19 traumatic stress, quality of life

## Abstract

The COVID-19 pandemic has brought new challenges and work changes for formal caregivers such as personal care aides with an impact on their quality of life (QoL). This cross-sectional study aims to analyze the relationships and contribution of sociodemographic and psychological variables towards QoL including the moderating role of self-care. This study included 127 formal caregivers from Portugal who were assessed on depression, anxiety and stress (DASS-21); professional self-care (SCAP); quality of life (SF-12); COVID-19 traumatic stress (COVID-19TSC) and preventive COVID-19 infection behaviors (PCOVID-19 IBS). Professional self-care was positively associated with QoL and also moderated the relationship between distress and QoL (*p* < 0.001). According to results, nursing homes should provide formal caregivers, such as personal care aides, with the professional support they need in order to promote their QoL and prevent burnout.

## 1. Introduction

Nowadays, the increase in adults 65 or above represents one of the main worldwide challenges with social and economic implications [1]. According to the United Nations [2], in 2019 there were 703 million people aged 65 or above. It is estimated that adults 65 or above will double to 1.5 billion by 2050. The inevitable increase in the number of older adults will cause a greater demand for services provided by formal caregivers carrying out the basic needs of daily living activities, such as assistance with food, hygiene and comfort, in residential care [3,4]. Formal caregivers such as personal support workers, health care aides and nursing aides [5,6] have not been the focus of research during the pandemic, yet they are on the front line by being in close contact with the patients they care for. As a result of their profession, formal caregivers in general report high levels of anxiety and depression [7,8] and low levels of QoL [9,10]. Furthermore, the duration of the care provided has been negatively associated with physical health [11] and QoL [12].

On March 11 of 2020, the World Health Organization (WHO, Geneva, Switzerland) [13] declared COVID-19, the disease caused by the new coronavirus, as a pandemic which led to the implementation of social distancing measures to prevent the spread of the virus [14] and also required nursing homes to close [14]. Formal caregivers had to face challenges every day to prevent the spread of COVID-19. In Portugal, during this time frame, there was a peak in the number of cases [15]. However, dependence on care or lack of Personal Protective Equipment (PPE) led to difficulties in complying with physical distancing [16]. In addition, the increased workload, fear of becoming infected and being infected, as well as the loss of patients and colleagues, increased the levels of stress related to COVID-19 in formal caregivers [17,18].

The pandemic impacted older adults’ behavior such as irritability, anxiety, apathy, agitation and sleep disorders that increased [19,20]. COVID-19 may be perceived as a traumatic event, since it created changes in lifestyle and interpersonal relationships, economic problems, uncertainty about the future, as well as fear of being infected and social distancing [21]. In formal caregivers, the pandemic was associated with new responsibilities and an overload of care that negatively impacted mental and physical health [22]. There is evidence that being a woman and having less work experience is associated with high levels of traumatic stress regarding COVID-19 [23,24]. According to the literature, social isolation and the COVID-19 pandemic impacted several aspects of QoL [25,26]. To prevent the spread of COVID-19, WHO implemented measures through preventive behaviors that included washing hands, avoiding crowds, wearing a mask and staying at home. Ayandele et al. [27] suggested that high levels of fear regarding COVID-19 were associated with increased preventive behaviors. Furthermore, preventive measures were shown to be negatively correlated with distress [28]. In the study by Geirdal et al. [29], social distancing showed a negative association with QoL.

High levels of distress in healthcare professionals due to COVID-19 have also been associated with limited resources, uncertain working conditions and long work hours, contributing to psychosocial risks associated with mental health such as depression, anxiety and trauma [30,31]. Lai et al. [32] found that 71.5% of formal caregivers (e.g., health professionals) exposed to COVID-19 reported high levels of distress. The literature also revealed that fear of becoming infected, being infected, lack of PPE and long shifts negatively impacted healthcare professionals and were associated with more distress [30,31]. In fact, the pandemic was associated with increased distress in caregivers [33] which contributed to a worse QoL [28].

Self-care is defined as a set of activities or strategies [34], health behaviors, professional development, professional support and work–life balance [35] that promotes mental and physical health [34]. The study by Ayala et al. [36] found that self-care played a moderating role between stress and QoL, in medical students. In addition, Pender et al. [37] found that self-care behaviors contributed to a better QoL and appeared to buffer the effect of stress on QoL [38]. Therefore, it seems relevant and important to study the role of self-care as a moderator between stress and QoL in formal caregivers in the face of the new demands and challenges due to the COVID-19 pandemic. The results will help to design new interventions to promote personal care aides’ QoL and decrease the negative impact of COVID-19.

The literature is consensual regarding the impact of the COVID-19 pandemic on QoL in formal caregivers. Some studies have addressed the impact of self-care on QoL in health professionals or psychologists, but they have focused on QoL related to work [39,40,41]. Additionally, studies evaluating self-care in formal caregivers have often included social support, physical exercise, or a healthy diet [42]; few have analyzed the moderating role of self-care in the relationship between stress and QoL [42]. The present study aims to fill this gap, considering the moderating role of self-care in the relationship between traumatic stress/distress in the face of COVID-19 and health-related QoL, as well as the relationship between those variables and the use of preventive behaviors against COVID-19 to improve QoL in formal caregivers.

In order to evaluate the contribution of sociodemographic and psychological factors on formal caregivers’ QoL, Pearlin’s Stress Process Model [43] was used as a reference. The model identifies background and contextual factors (e.g., age, gender) that interfere with primary stressors (e.g. patient’s characteristics; care situation, needs of the caregiver) and secondary stressors (e.g. work interference, financial strain) and the caregiver’s appraisal considered as the caregiver’s subjective assessment of their role (e.g. adequacy of resources; perceived control), contributing to the outcome (physical and psychological health). The model also predicts moderating variables between the two types of stressors, appraisal and the outcome that may mitigate or exacerbate the impact of stress on the caregiver’s health such as coping strategies, social support, self-efficacy, personality and other exacerbating or ameliorating factors. In the present study, background variables assessed included age, gender, education, duration of care, number of working hours per day and number of days off. Traumatic stress regarding COVID-19 was considered a primary stressor impacting preventive behaviors against COVID-19 in the work environment as well as distress, which were considered secondary stressors. QoL was considered the main outcome in its physical and psychological dimensions. Self-care behaviors (autoregulation strategy) were considered the moderating variable between the stressors and the outcome since they have the potential to mitigate the impact of stressors on QoL. Appraisal was not assessed in the present study.

The specific aims of this study were: (i) to explore the differences on QoL according to caregiver’s contextual variables; (ii) to find the variables that contributed to QoL; and (iii) to analyze the moderating role of self-care in the relationship between COVID-19 traumatic stress/distress and QoL. Based on this model, the following hypotheses were formulated: H1: a shorter duration of care, less traumatic stress in the face of COVID-19, less distress, more self-care behaviors and less preventive behaviors against COVID-19 infection will be positively associated with better QoL; H2: duration of care, traumatic stress in the face of COVID-19, self-care, and preventive behaviors against COVID-19 infection and distress will predict QoL; and H3: self-care is expected to moderate the relationship between traumatic stress regarding COVID-19 and QoL as well as in the relationship between distress and QoL.

## 2. Materials and Methods

### 2.1. Study Design, Participants and Procedure

The study used a cross-sectional design. Data collection was carried out in a face-to-face format and included the caregivers of the five private institutions of social solidarity in the north of Portugal. The population included all the nursing homes from a district in the north of Portugal. The total population was 135 aides. All were invited to participate but only 127 agreed to participate. Inclusion criteria included being a personal care aide; exercising the role of a formal caregiver in a nursing home or providing home care for at least 1 year; and being over 18 years old. Exclusion criteria included receiving psychological or psychiatric support.

The present study was approved by the Ethics Committee for Research in Social and Human Sciences of the university where the researchers belonged (CEICSH/030-2021). The first step of the protocol was to contact the nursing homes and explain the purpose of the study. Data were collected face-to-face in nursing homes by two psychologists. All formal caregivers were told the purpose of the study and assured of the confidentiality and anonymity of the data, as well as the voluntary nature of participation. All participants signed an informed consent form prior to answer the questionnaires. Data collection took place from December 2021 to February 2022.

### 2.2. Measures

#### 2.2.1. Sociodemographic Questionnaire [44]

This instrument consists of 10 items that assessed the following sociodemographic variables: age, sex, marital status, education level, caregiver typology, duration of care, number of working hours per day, number of weekly days off and whether caregivers received psychological or psychiatric support (self-report).

#### 2.2.2. Short-Form Health Survey-12 (SF-12) [45,46]

The SF-12 scale is an instrument that assesses health-related QoL, developed from the Short-Form Health Survey-36 Scale (SF-36). It consists of 12 items and is composed of two health dimensions: General Physical Health and General Mental Health. The General Physical Health dimension includes physical function, physical performance and general health. The General Mental Health dimension is composed of mental health, emotional performance, social function and vitality. The total score ranges from 0 to 100, with a higher score indicating better QoL. Caregivers’ answers were made on a 5-point Likert scale. The original scale has a Cronbach’s alpha of 0.89 for the General Physical Health dimension and 0.76 for the General Mental Health dimension. The Portuguese version has an alpha of 0.85 for the full scale [46]. In the present study, Cronbach’s alpha for the full scale was 0.85.

#### 2.2.3. Depression, Anxiety and Stress Scale (DASS-21) [47,48] 

The scale assesses levels of depression, anxiety and stress. It is composed of 21 items and includes three subscales: depression, anxiety and stress. Each item is answered on a 4-point Likert scale (“0” corresponds to “did not apply to me at all” and 3 to “applied to me a few times”). Participants replied to the items considering the previous week. The original version has a Cronbach’s alpha of 0.81 for the depression subscale, 0.89 for the anxiety subscale and 0.78 for the depression subscale. In the Portuguese version [48], the alpha was 0.85 on the depression scale, 0.74 for anxiety and 0.81 for stress, and for the total scale was 0.94 [49]. In the present study, Cronbach’s alpha was 0.83 for the depression subscale, 0.82 for the anxiety subscale and 0.87 for the stress subscale, with an alpha of 0.93 for the global scale.

#### 2.2.4. Self-Care Assessment for Psychologists Scale (SCAP) [35,50]

The questionnaire assesses general self-care behaviors such as: “I share positive work experiences with colleagues” and “I spend time with family or friends”. Since the questions may be applied to any healthcare professional, this instrument was used for the formal caregivers in the present study. The scale includes 21 items with 5 dimensions: Professional Support, Professional Development, Life Balance, Cognitive Strategies and Daily Balance. Each item is answered on a 7-point Likert scale (1—never, 7—almost always). Cronbach’s alphas for the original version were 0.83 (Professional Support), 0.80 (Professional Development), 0.81 (Life Balance), 0.72 (Cognitive Strategies) and 0.70 (Daily Balance) for each dimension [49]. The Portuguese version [50] showed a Cronbach’s alpha of 0.86 for Professional Support, 0.83 for Professional Development, 0.88 for Life Balance, 0.84 for Cognitive Strategies and 0.70 for Daily Balance. In the present study, the alpha was 0.75 for the Professional Support dimension, 0.82 for Professional Development, 0.84 for Life Balance, 0.78 for Cognitive Strategies and 0.47 for Daily Balance. Due to the low alpha, the Daily Balance subscale was not used in the statistical analyses.

#### 2.2.5. Preventive COVID-19 Infection Behaviors Scale (PCOVID-19IBS) [51,52]

This instrument was developed according to the preventive behaviors recommended by the WHO, consisting of 5 items, answered on a 5-point Likert scale, where “1” corresponds to “almost never” and “5” to “almost always”. A higher score indicates that preventive behaviors are performed more frequently. Regarding internal consistency, the alpha was 0.82 in the original version and 0.62, in the Portuguese version [52]. In the present study, Cronbach’s alpha was 0.65. Considering the number of items (only 5), the alpha may be considered adequate [53].

#### 2.2.6. COVID-19 Traumatic Stress Scale (COVID-19TSC) [52,54]

This instrument assesses traumatic stress regarding COVID-19 through 12 items and includes three subscales: threat/fear of present and future infection and death subscale, economic trauma subscale, and routine disorder and isolation subscale. Questions are answered on a 5-point Likert scale (0—never, 4—often). Higher scores indicate more trauma due to COVID-19. The original version has a Cronbach’s global alpha of 0.88. The Portuguese version [52] showed a global alpha of 0.77. In the present study, the alpha was 0.75 for the global scale.

### 2.3. Data Analysis

The data were analyzed using the IBM SPSS^®^ program (Statistical Package for the Social Sciences) version 28.0. Considering a power level of 0.80, a moderate effect size of 0.1, a probability level of 0.05 and six predictors, the sample size required was 97 participants. Descriptive statistics were used to characterize the sample (calculation of frequencies, means and standard deviations). As the assumptions for the use of parametric tests were met, Pearson’s Correlation (H1) tests were performed to calculate the relationships between the variables.

In order to test the variables that contribute to QoL, a Multiple Linear Regression was used, since all the assumptions for parametric test were fulfilled. The variables with the highest correlation with QoL were included in the regression model. In the first model, the sociodemographic variable: duration of care was introduced, and in the second model, the variables: distress, COVID-19 traumatic stress, professional support, life balance and cognitive strategies were added (H2).

Finally, to test the moderating role of self-care behavior between COVID-19 traumatic stress and QoL, as well as between distress and QoL, the Macro PROCESS for SPSS version 3.5 and the Johnson-Neymar (JN) technique were used.

## 3. Results

### 3.1. Sample Description

The sample included 127 formal caregivers, all women who provided care and support in nursing homes/older adults’ homes, aged between 21 and 66 years of age (M: 42.2, SD: 10.1). All the formal caregivers were responsible for the daily living activities of older adults. Concerning education, the average was 11.4 (SD = 3.4). Caregivers’ sociodemographic characterization is shown in Table 1.

### 3.2. Differences in QoL According to Caregiving Contextual Variables

Regarding sociodemographic variables, the results showed a negative association between duration of care and QoL (r = −0.20, *p* < 0.05). Therefore, the longer the duration of care, the worse the QoL. Regarding the psychological variables, the results revealed a negative association between distress (r = −0.54, *p* ≤ 0.001), COVID-19 traumatic stress (r = −0.43, *p* ≤ 0.001) and the QoL. Therefore, higher levels of distress and COVID-19 traumatic stress were associated with a worse QoL. The results also revealed that professional support (r = 0.37, *p* ≤ 0.001), professional development (r = 0.25, *p* ≤ 0.01), life balance (r = 0.38, *p* < 0.001) and cognitive strategies (r = 0.47, *p* ≤ 0.001) were positively associated with QoL. Therefore, high levels of professional support, professional development, life balance and cognitive strategies were associated with better QoL. Age (r = −0.12, *p* = 0.192), number of years of education (r = 0.06, *p* = 0.521) and preventive behaviors against COVID-19 infection (r = −0.04, *p* = 0.658) did not correlate with QoL. The results of the correlations are represented in Table 2.

### 3.3. Contributors to QoL

Preliminary analyses included correlations between the variables (H2) presented in Table 2. Model 1 evaluated the contribution of sociodemographic variables to QoL, revealing that the duration of care explained 4% of the total variance, with the model being significant ($R^{2}=$ 0.04, F (1, 125) = 5.25, *p* = 0.024) and the duration of care provided contributed significantly to QoL (β = −0.22, t = −2.29, *p* = 0.024). When psychological variables were added, model 2 explained 49% of the total variance ($R^{2}=$ 0.49, F (5, 120) = 20.95, *p* < 0.001). Therefore, the final model showed duration of care (β = −0.17, t = −2.20, *p* = 0.030), distress (β = −0.24, t = −4.49, *p* < 0.001), traumatic stress due to COVID-19 (β = −0.31, t = −3.95, *p* < 0.001) and cognitive strategies (β = 0.35, t = 2.06, *p* = 0.042) to significantly contribute to formal caregivers’ QoL. However, professional support (β = 0.10, t = 0.84, *p* > 0.05) and life balance (β = 0.20, t = 1.36, *p* > 0.05) did not contribute to QoL. The regression results are shown in Table 3.

### 3.4. Moderating Role of Self-Care between COVID-19 Traumatic Stress and QoL

When testing the moderating role of self-care between traumatic stress regarding COVID-19 and QoL, no significant results were found. Therefore, professional support was not a moderator between traumatic stress regarding COVID-19 and QoL (β = 0.01, 95% CI [−0.02, 0.04], t = 0.74, *p* = 0.4592). The same was true for professional development (β = 0.01, 95% CI [−0.03, 0.04], t = 0.31, *p* = 0.7607), life balance (β = −0.01, 95% CI [−0.04, 0.03], t = −0.29, *p* = 0.7695) and cognitive strategies (β = 0.01, 95% CI [−0.04, 0.06], t = 0.44, *p* = 0.6587) that did not play a moderating role between traumatic stress regarding COVID-19 and QoL.

### 3.5. Moderating Role of Self-Care between Distress and QoL

The model that tested the moderating role of self-care in terms of professional support between distress and QoL was significant, F (3, 123) = 23.90, *p* < 0.001, β = −0.03, 95% CI [−0.05; −0.00], *p* = 0.0260, explaining 36.51% of the variance. Therefore, the negative relationship between distress and QoL is less strong when the formal caregiver receives more professional support, β = −0.54, 95% CI [−0.74; −0.33], t = −5.16, *p* ≤ 0.0000. The Johnson-Neyman (JN) Technique revealed that distress was significantly correlated with QoL when the standardized value of professional support was −8.09 (β = –0.17, *p* = 0.05), corresponding to 89.7% of the sample (Figure 1). However, the other subscales of self-care, such as professional development (β = 0.01, 95% CI [−0.01, 0.02], t = 0.79, *p* = 0.4339), life balance (β = 0.01, 95% CI [−0.01, 0.03], t = 0.84, *p* = 0.4000) and cognitive strategies (β = −0.01, 95% CI [−0.04, 0.01], t = −0.93, *p* = 0.3540), did not moderate the relationship between distress and QoL.

## 4. Discussion

This study examined the associations between QoL and sociodemographic and psychological variables in formal caregivers caring for the daily activities of older adults with dementia. The results revealed that the duration of the care provided was negatively correlated with QoL. This result matches the literature, since studies indicate that caregivers and health professionals with more years of care provided report worse QoL [12,55,56].

Distress levels were also negatively correlated with QoL. There are studies that corroborate these results [57,58]. Caregivers have high levels of distress due to negative physical, emotional and social experiences [59]. These high levels of distress have been associated with a worse QoL [60]. During the pandemic, concerns about the risk of infection, as well as concerns about PPE, contributed significantly to an increase in distress and a decrease in QoL [58].

As expected in this study, COVID-19 traumatic stress was negatively correlated with QoL. Several studies corroborate this result [61,62]. Formal caregivers had to undergo changes at their work including working conditions, as well as new responsibilities due to the pandemic, with negative consequences on their mental health [22].

The results of the present study also revealed that preventive behaviors against COVID-19 infection did not correlate with QoL. Social distancing was found to be negatively associated with QoL [29]. However, the research by Zhou and Zhang [41] revealed that hand hygiene did not correlate with health professionals’ mental health but was negatively correlated with professional QoL. Furthermore, the difficulties that arise in complying with preventive behaviors, such as social distancing and hygiene, were negatively associated with QoL [63]. In this study, the focus was on how often professionals practiced preventive behaviors and not on the difficulties in complying with preventive behaviors, which may explain the results that were found. In addition, self-care correlated positively with QoL, which is in line with the literature that revealed that more self-care behaviors in psychologists were positively associated with QoL [39].

In this study, a longer duration of care contributed negatively to QoL. In a study by Kheiraoui et al. [55], duration of care was a predictor of QoL. However, in the study by Muthuri et al. [56], the duration of care did not contribute to QoL. This inconsistency may be explained considering that during the pandemic, caregivers with longer duration of care had to assume the coordination and control of services in institutions [64], since those professionals with more years of experience were called upon often to make decisions in the workplace to manage the provision of care [65].

The results revealed that distress contributed negatively to QoL. These results are in accordance with the literature [28,66]. In fact, in studies of formal caregivers, anxiety and depression have significantly contributed to QoL [67]. Furthermore, distress was found to negatively contribute to QoL in the general population during the pandemic [28]. In turn, COVID-19 traumatic stress contributed negatively to QoL. These results are in line with the literature [61,62]. The risk of contracting COVID-19 contributed negatively to QoL in healthcare professionals [62]. Moreover, increased working hours, social isolation, fear of being infected or getting infected, loss of patients and colleagues [17,18,61] and concerns about the lack of PPE [61] contributed to a worse QoL [61]. However, cognitive self-care strategies contributed positively to QoL. Therefore, formal caregivers with more use of cognitive strategies reported a better QoL. These results match other studies [39,61]. In fact, the results of Li et al. [61] found that coping styles contributed to better QoL in healthcare professionals during the COVID-19 pandemic.

Contrary to what was predicted, according to Pearlin et al.’s model [42], and the literature [36,68], it was not possible to assess the moderating role of self-care between traumatic stress regarding COVID-19 and QoL. In fact, the results of the study by Ayala et al. [39] found that self-care was not a moderator between stress and QoL, in psychologists.

The results found that professional support, one of the self-care subscales, was a moderator between distress and QoL. The literature found a direct relationship between distress and QoL, with high levels of distress being associated with worse QoL [28,66]. Furthermore, the moderating role of professional support in this relationship makes intuitive sense, given that professionals with more professional support revealed better professional QoL [69].

The results of the present study corroborate the Stress Process Model by Pearlin et al. [43], and the model was shown to be adequate to this sample. Additionally, the moderating role of professional support between distress and QoL was confirmed, as well as the impact of the duration of care, distress, traumatic stress concerning COVID-19 and cognitive strategies on formal caregivers’ QoL.

### Limitations and Future Implications

The present study has some limitations that should be acknowledged, such as the cross-sectional design that does not allow for cause-and-effect relationships, as well as the use of self-report instruments and the fact that a convenient sample was used collected in one single district of Portugal. In addition, the number of formal caregivers in each caregiver typology was very unequal, which did not allow for comparison between the different typologies of caregivers regarding QoL. The same was true for the number of weekly days off and the working hours per day of the participants. Similarly, the inequality of the gender representativeness of the sample is a limitation and should be addressed in future research. Future studies should employ bigger samples and include appraisal variables in order to study if they play a mediating role in the relationship between primary/secondary stressors and QoL.

Future research should also include a longitudinal design and analyze the impact of psychological variables on personal care aides’ QoL, over time. Moreover, it would also be important to implement qualitative studies to understand formal caregivers’ experience, in order to get more in-depth and detailed information about their needs, since studies in this population of formal caregivers are very much neglected in the literature.

## 5. Conclusions

Based on the results of the present study, it is important to highlight that longer duration of the care as well as traumatic stress regarding COVID-19 and distress contributed to a worse QoL, while self-care contributed positively to the QoL of formal caregivers. The moderating role of the self-care professional support subscale in the relationship between distress and QoL emphasizes the importance of professional support during the COVID-19 pandemic.

Considering that in many countries, COVID-19 is still a pandemic, nursing homes should provide formal caregivers with the professional self-care support they need, in order to promote their QoL and prevent burnout. Such professional support may include intervention programs that promote professional relationships as well as the sharing of stressful work situations and positive experiences, and a solid work support network that results in reducing isolation and will certainly be important in decreasing distress and increasing QoL. In conclusion, the present study highlights the importance of self-care in formal caregivers such as personal care aides.

## Figures and Tables

**Figure 1 ijerph-20-05177-f001:**
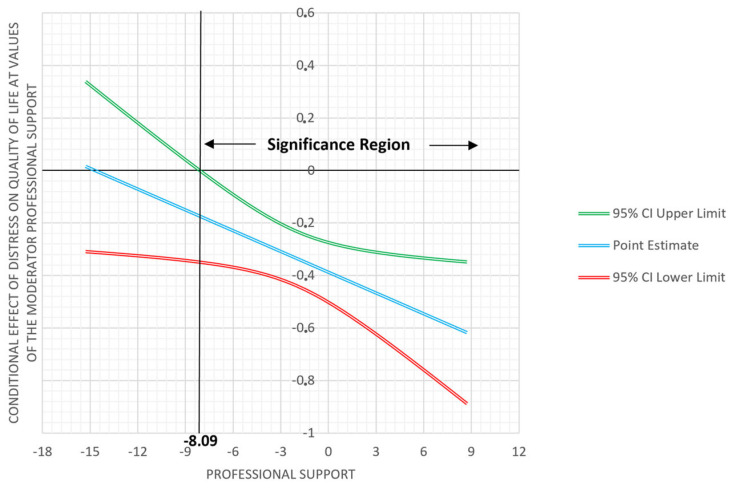
Professional Support as a moderator in the relationship between distress and QoL.

**Table 1 ijerph-20-05177-t001:** Sociodemographic characterization of caregivers (N = 127).

Continuous Variable	Min	Max	Mean	SD
Age	21	66	42.2	10.1
Education (years)	4	20	11.4	3.4
Duration of care (in years)	1	32	7.8	6.8
Categorical Variables	Frequency		%	
Material Status				
Single/Divorced/Separated	42		33.1	
Married	85		66.9	
Caregiver Typology				
Formal caregiver—Nursing home	100		78.7	
Formal caregiver—Home Support	27		21.3	
Working hours per day				
0–8 h	113		89.2	
8–18 h	14		10.8	
Weekly days off				
None	2		1.6	
1 day off	12		9.4	
2 days off	107		84.3	
More than two	6		4.7	

**Table 2 ijerph-20-05177-t002:** Relationship between all variables.

Measures	1	2	3	4	5	6	7	8	9	10	11	12	13
1. Quality of Life	—												
2. General Physical Health Dimension	0.87 ***	—											
3. General Mental Health Dimension	0.85 ***	0.54 ***	—										
4. Age	−0.12	−0.28 **	0.07	—									
5. Duration of care	−0.20 *	−0.21 *	−0.16	0.25 **	—								
6. Years of education	0.06	0.19 *	−0.09	−0.39 ***	−0.17	—							
7. Distress	−0.54 ***	−0.42 ***	−0.45 ***	0.01	0.12	−0.11	—						
8. COVID-19 traumatic stress	−0.43 ***	−0.30 ***	−0.37 ***	−0.01	0.09	0.02	0.30 ***	—					
9. Preventive behaviors	−0.04	0.06	−0.10	−0.09	0.04	0.05	0.03	0.47 ***	—				
Self-Care													
10. Professional Support	0.37 ***	0.29 ***	0.36 ***	−0.02	0.16	0.23 *	−0.30 ***	−0.14	0.04	—			
11. Professional Development	0.25 **	0.20 *	0.31 ***	−0.10	0.17	0.12	−0.12	−0.03	0.03	0.55 ***	—		
12. Life Balance	0.38 ***	0.24 **	0.43 ***	−0.15	0.10	0.29 **	−0.23 *	−0.10	0.09	0.57 ***	0.64 ***	—	
13. Cognitive Strategies	0.47 ***	0.35 ***	0.47 ***	−0.15	−0.05	0.23 **	−0.34 ***	−0.12	0.10	0.57 ***	0.47 ***	0.65 ***	—

* *p* < 0.05; ** *p* < 0.01; *** *p* < 0.001.

**Table 3 ijerph-20-05177-t003:** Variables that contributed to QoL.

Variables	Model 1		Model 2	
	*B*	*t*	*B*	*t*
-Duration of care	−0.22 *	−2.29 *	−0.17 *	−2.20 *
-Distress			−0.24 ***	−4.49 ***
-COVID-19 Traumatic Stress			−0.31 ***	−3.95 ***
-Professional Support			0.10	0.84
-Life Balance			0.20	1.36
-Cognitive Strategies			0.35 *	2.06 *
R^2^	0.04		0.49	
F	5.25 *		19.03 ***	
∆R^2^	0.03		0.45	
∆F	5.25 *		20.95 ***	

* *p* < 0.05; *** *p* < 0.001.

## Data Availability

The data presented in this study are available on request from the corresponding author.

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
