# Peer review of "The Moderating Role of Self-Care Behaviors in Personal Care Aides of Older Adults during the COVID-19 Pandemic"

_ijerph, 2023, doi:10.3390/ijerph20065177_

Round 1

Reviewer 1 Report

Thank you for the opportunity to read this very interesting paper. The paper has many strengths that should be of interest to the journal audience. Thus, the following suggestions are around enhancing the presentation for publication and clarifying aspects of the data and reporting.

I will go by line number for the most part. If not, I will try to be as specific as possible in noting the area I am speaking about.

Abstract

Purpose. It is necessary to include the country.

Where are p-values? (p < .001). Authors must specify it. P values showing the differences between groups should be given.

Keywords. Psychological Morbidity; Covid-19 Preventive Behaviors; Covid-19 Traumat-20 ic Stressare not MeSH Terms.

 1. Introduction

Authors must speak more about gender and professionals healthcare, about the adverse working conditions affecting the mental health and QoL, which are the consequences of high levels of psychological distress (professionals and patients), etc. Are there in these environments a high presence of symptomatology related to work stress (physical and emotional fatigue, overload, tension, and anxiety) that may pose a risk of impaired mental health? Why?

The prevalence and incidence of COVID-19 in the area of study during the period of study should be discussed.

2. Materials and Methods

[125] I suggest including Table I in the Results section

Do the authors have a study protocol? The study protocol should be described in detail. How was the sample chosen? Did the authors calculate the needed sample size? Please, clarify. How was the sample size determined? Did the authors test power calculation? Authors must specify it.

How was the sample chosen? Which is the total population? Authors must specify it.

STATISTICAL ANALYSIS: it is convenient to run, and describe, the analysis of normality in the distribution of scores of the validated questionnaires they have used

3. Results

Tables. Please, provide the information about ***

 At last, but not least, I recommend you to make available your data in an open repository. I think it will make this scientific process more transparent, and it allows other researchers to replicate your results.

4. Discussion

Limitations related with the type of methodology used. Limitations regarding representativeness of respondents should be better addressed Authors must specify it. The fact of having a convenience sample should be included in the limitations of the study.

I wish you all the best.

Author Response

We would like to thank you for your help in this process, as well as the anonymous reviewers for their valuable insights and comments, which have greatly contributed to improve the quality of our manuscript.

Reviewer 1

Thank you for the opportunity to read this very interesting paper. The paper has many strengths that should be of interest to the journal audience. Thus, the following suggestions are around enhancing the presentation for publication and clarifying aspects of the data and reporting. I will go by line number for the most part. If not, I will try to be as specific as possible in noting the area I am speaking about.

Abstract

Purpose. It is necessary to include the country.

Answer: The country was included (line 17).

Where are p-values? (p < .001). Authors must specify it. P values showing the differences between groups should be given.

Answer: The p values were added (lines 20).

Keywords. Psychological Morbidity; Covid-19 Preventive Behaviors; Covid-19 Traumat-20 ic Stressare not MeSH Terms.

Answer: The key words are not MeSH Terms. Self-care, Formal caregivers, Psychological Morbidity, Covid-19 Preventive Behaviors, Covid-19 Traumatic Stress; Quality of Life.

  1. Introduction

Authors must speak more about gender and professionals healthcare, about the adverse working conditions affecting the mental health and QoL, which are the consequences of high levels of psychological distress (professionals and patients), etc. Are there in these environments a high presence of symptomatology related to work stress (physical and emotional fatigue, overload, tension, and anxiety) that may pose a risk of impaired mental health? Why?

Answer: Information regarding gender and professional health care was added (lines 52-54, and 62-65).

The prevalence and incidence of COVID-19 in the area of study during the period of study should be discussed.

Answer: There were no studies that analysed the prevalence of covid-19 in this population. This is the first study focused on personal care aides. In Portugal, during this time frame, there was a peak in the number of cases. That information was added in the paper ( line 42-44).

  1. Materials and Methods

[125] I suggest including Table I in the Results section

Answer: Table I was moved to the results section with the sample description (lines 223-229).

Do the authors have a study protocol? The study protocol should be described in detail. How was the sample chosen? Did the authors calculate the needed sample size? Please, clarify. How was the sample size determined? Did the authors test power calculation? Authors must specify it.

Answer: The first step of the protocol was to contact the nursing homes and discuss and explain the purpose of the study. After the signature of the consent form, the study followed a face-to-face interview where each caregiver answered the questionnaires in residence care (lines 135-136).

The total population included 135 aides who were invited but only 127 agreed to participate. This information was added in the paper (lines 127-129).

The sample size was determined using a sample size calculator for multiple regression from Daniel Sopper. Considering a power level of .80, a moderate effect size of 0.15, a probability level of .05, and 6 predictors, the sample size required was 97.

[Soper, D.S. (2023). A-priori Sample Size Calculator for Multiple Regression, https://www.danielsoper.com/statcalc].

STATISTICAL ANALYSIS: it is convenient to run, and describe, the analysis of normality in the distribution of scores of the validated questionnaires they have used

Answer: The variables in the validated questionnaires followed a normal distribution according to Kolmogorov-Smirnov (K-S) test. Nonetheless, with large enough sample sizes, as in this study; following the central limit theorem, the violation of the normality does not cause major problems and in fact parametric procedures should be used in samples over 100 participants.

 [Mishra, P., Pandey, C. M., Singh, U., Gupta, A., Sahu, C., & Keshri, A. (2019). Descriptive statistics and normality tests for statistical data. Annals of cardiac anaesthesia, 22(1), 67.]

  1. Results

Tables. Please, provide the information about ***

Answer: The meaning of *** was provided for all tables.

 At last, but not least, I recommend you to make available your data in an open repository. I think it will make this scientific process more transparent, and it allows other researchers to replicate your results.

Answer: Thank you for the suggestion. At present the data is available upon request and at the end of the larger research project, where this study is part of, it will be openly available.

  1. Discussion

Limitations related to the type of methodology used. Limitations regarding representativeness of respondents should be better addressed Authors must specify it. The fact of having a convenience sample should be included in the limitations of the study.

Answer: That information was added (lines 358, 362-365).

Reviewer 2 Report

Thank you for the opportunity to revise the paper "The Moderating Role of Self-Care Behaviors in Personal Care Aides of Elderly People During the COVID-19".

The paper is interesting and well organized. Some minor improvements can be made.

please clarify sentence in lines 89-92. 

check the sentence in lines 209-211.

typos: car in line 350

Author Response

Reviewer 2: Thank you for the opportunity to revise the paper "The Moderating Role of Self-Care Behaviors in Personal Care Aides of Elderly People During the COVID-19". The paper is interesting and well organized. Some minor improvements can be made.

please clarify sentence in lines 89-92. 

Answer: The sentence was changed and clarified (lines 92-95).

Answer: The sentence was changed and clarified (lines 213-218).

typos: car in line 350

Answer: The word “car” was changed to “care” in line 367.

Reviewer 3 Report

This is an informative manuscript that describes the moderating role of self-care behaviors in personal care aids of older adults during the COVID-19 pandemic.  Thank you for addressing an important issue-self-care in formal caregivers. I offer some feedback I hope is helpful.

Introduction:

-What is meant by “behavioral impairment” in line 46?

-Purpose/aims of study are clearly presented.

 Methods:

- Please clarify if the participants were only recruited from nursing homes? Or did some provide home support as well?

-Consider adding duration of data collection, including months/year data was collected

-How was receiving psychological or psychiatric support measured? Self-report? (line 128)

-Please provide significance for p-value in data analysis section

Results: Clearly presented.

-Recommend adding significance for * values as footnote at the bottom of Table 2 and Table 3.  

Discussion:

-Recommend adding the sample was all women in limitations section.

Minor Comments: 

-Consider changing “elderly” to older adults throughout the manuscript and title.

-Consider changing “ageing population” to adults 65 or above or persons 65 years and older throughout the manuscript.

- Please conduct a thorough check of for typos throughout the manuscript (“include” in line 13 should be including, “it” should be inserted in line 47, “Found” in line 321 should be lowercase, “car” should be care in line 350)

Author Response

Reviewer 3: This is an informative manuscript that describes the moderating role of self-care behaviors in personal care aids of older adults during the COVID-19 pandemic.  Thank you for addressing an important issue-self-care in formal caregivers. I offer some feedback I hope is helpful.

Introduction:

-What is meant by “behavioral impairment” in line 46?

Answer: The sentence was clarified (lines 47-8).

-Purpose/aims of study are clearly presented.

Answer: Thank you.

 Methods:

- Please clarify if the participants were only recruited from nursing homes? Or did some provide home support as well?

Answer:  As shown in Table 1, most participants were formal caregivers from nursing homes     (100), and only 27 were formal caregiver providing home support.

-Consider adding duration of data collection, including months/year data was collected

Answer: Data collection took place from December 2021 till February 2022) . That information was added in the paper (lines 139 – 140).

-How was receiving psychological or psychiatric support measured? Self-report?

Answer: That information was collected in the sociodemographic questionnaire (participants’ self-report). That information was added in the manuscript (lines 145-146)

- Please conduct a thorough check of for typos throughout the manuscript (“include” in line 13 should be including, “it” should be inserted in line 47, “Found” in line 321 should be lowercase, “car” should be care in line 350)

Answer: The paper was revised by a native English speaker. Those typos were corrected (lines 16, 49, 338, 367, respectively).